# A Survey and Tutorial on Network Optimization for Intelligent Transport System Using the Internet of Vehicles

**DOI:** 10.3390/s23010555

**Published:** 2023-01-03

**Authors:** Saroj Kumar Panigrahy, Harika Emany

**Affiliations:** 1School of Computer Science and Engineering, VIT-AP University, Near Vijayawada 522237, Andhra Pradesh, India; 2Delhi Public School, Nacharam, Hyderabad 500076, Telengana State, India

**Keywords:** intelligent transport system (ITS), internet of things (IoT), internet of vehicles (IoV), vehicular ad hoc network (VANET), network optimization

## Abstract

The Internet of Things (IoT) has risen from ubiquitous computing to the Internet itself. Internet of vehicles (IoV) is the next emerging trend in IoT. We can build intelligent transportation systems (ITS) using IoV. However, overheads are imposed on IoV network due to a massive quantity of information being transferred from the devices connected in IoV. One such overhead is the network connection between the units of an IoV. To make an efficient ITS using IoV, optimization of network connectivity is required. A survey on network optimization in IoT and IoV is presented in this study. It also highlights the backdrop of IoT and IoV. This includes the applications, such as ITS with comparison to different advancements, optimization of the network, IoT discussions, along with categorization of algorithms. Some of the simulation tools are also explained which will help the research community to use those tools for pursuing research in IoV.

## 1. Introduction

The Internet of Things (IoT), through which a sizable amount of physical gadgets are connected with the web, has risen from ubiquitous computing and the Internet itself. To make efficient use of the available network, it is important to provide solutions to various network-related IoT problems, including routing, congestion, quality of service (QoS), heterogeneity, energy conservation, scalability, reliability, and protection. This paper presents a comprehensive survey on network optimization in IoT. It highlights the background of IoT and IoV. This study also discusses about the applications of IoV which include Intelligent Transport System (ITS) with comparison to different advancements, optimization of the network along with categorization of various algorithms.

### 1.1. Intelligent Transport System

The ITS, introduced for resolving transportation problems and improving overall effectiveness of transportation. Within the framework of smart cities, the ITS is subject to smart mobility, which has been gaining popularity in recent decades. Hall et al. [1] proposed that a smart city should keep track of its components (e.g., roads, buildings, etc.) to optimise its services to the best of its capabilities, plan maintenance activities which are preventive, and security monitoring while expanding utilities for denizens. A smart car is a crucial component of IoT, an application of ITS. It has access to the web, shares information with other smart devices that are inside and outside the car.

The CMSWire predicts that over 380 M vehicles are expected to be on the roads by 2020 [2]. The Business Insider anticipated 94 M connected vehicles by 2021 and 82% will be connected to other cars, traffic lights, road side units (RSU), etc. [3]. Technologies such as artificial intelligence, big data, and machine learning shall work towards detecting deterrents on roads, or unfavourable climate, to bring down road congestions and traffic mishaps in the near future. Advanced data mining techniques can give rise to new revenue marketing which is based on individual driver’s visited locations, vehicular entertainment, and content choice and can also be a new source of earnings [4]. Different applications include saved driver data profiles that will allow us to create personalized driving experiences and navigation. When it senses the person’s presence near the car, smart infotainment plays the driver’s favourite music. Radios and CD players are now being replaced with ‘Smart Infotainment Devices’ and there are infinite possibilities. Leading manufacturers of cars, such as British Motor Works, Volkswagen, Nissan, Porsche, Audi, Mercedes Benz, Tesla, and Jaguar, are promoting the Internet of Vehicle (IoV) [5]. Car manufacturers are already creating major advancements in the trials of vehicular technology. One such case, Telstra, which is in collaboration with Cohda Wireless, effectively ran Vehicle to Person (V2P) technology trials in South Australia Auto [6]. The technology was tested using everyday scenarios, such as a pedestrian advancing towards a blind curve. Numerous relevant platforms already exist to discover ingenious methods to motivate separate developers to assist in building connected car environments (such as Android Auto, Baidu CarLife, MirrorLink, Apple car play, etc.) that provide infotainment on smartphones. Google’s self-driving automobile venture is most likely the one with major headway. More than five million miles of street testing has already been conducted by them. Smart technology that is helping to reduce traffic congestion has already been equipped in 50 intersections in Pittsburgh, Pennsylvania. As a component of its smart mobility project, the Austin transportation division which exists in the United States has just tried experimenting with smart parking meters.

### 1.2. Motivation

The motivation behind the architecture and advancement of IoV is split into three sections.
The commercialization issues in vehicular ad hoc networks (VANET);Traffic issues;Market opportunities.

#### 1.2.1. The Commercialization Issues in VANET

The VANET by itself was unable to ensure international and imperishable services through ITS applications. Some of the causes of commercial issues in VANET are pinpointed below. This is due to the ad hoc network nature and dynamic networks where some vehicles fall out of communication range and network region, lose its services from the network despite being on-road. In VANET, the Internet is not completely guaranteed, drivers and passengers are not able to receive commercial applications despite the substantial bloom of personal gadgets, the gadgets are unable to interact with VANET because of the incompatible architecture of the network in present VANET [7]. This issue can be attributed to the limitations on computing and storage and the non-availability of services in automobiles [8].

#### 1.2.2. Traffic Issues

Safety, efficiency, and pollution are related to on-road traffic which are root of concern on the design and advancement of IoV. Reliable communication offered by ITS would successfully bring down congestion casualties [9]. The increasing number of traffic casualties around the globe has been mentioned in many surveys [10]. The World Health Organization (WHO) reports that as of now, road accidents cause almost 1.24 million deaths around the globe [11], by 2030, road accidents could represent up to 40% of all deaths [12,13]. It was reported in [14] that the count between the time of an early cautionary alert and the motorist doing something about it ranges between three-fourths to one and a half of a second. People between the ages of 15 and 44 form a major share of worldwide road deaths (about 59%) as per another report. In New Delhi, India, fuel valued at over USD 1.6 million is wasted daily, due to vehicles idling in road congestion [14]. Traffic can lead to major drawbacks, such as fuel, time, economic wastage, and environmental pollution. It gives way to drastic repercussions on travel, regional and national economies, enterprises, and people. Each year, nearly 227 hours are spent in traffic by London drivers [15]. The reports convey that there is an urgent need to bring down casualties on roads. For these reasons, the usage of more dependable vehicular communication for safety applications is required.

#### 1.2.3. Market Opportunities

The automobile industry and many more industries, including the software industry, IT equipment manufacturers, and web access providers, were offered a market opportunity in IoV. Autonomous vehicles are usually equipped with over 300 sensors with the capability of producing more than 5000 GB of data daily [16]. Generated historical data are utilized for making subsequent choices, such as forecasting congestion levels and determination of an optimal route. There are almost three million electric vehicles in 2017 as compared to the 0.7 million vehicles in 2014 [17]. Electric vehicles are not a dream any more but are a fact of life  [18]. More of these vehicles are expected on the roads with each passing year. More than five million miles of road trials have been conducted by Google on its self-driving automobiles. Tesla’s Model S, which was introduced to the market in 2012, led the connected vehicles market in that period. The smart car sale could go up to nearly 81 million per year; the latest cars are expected to have a form of connected drive technologies in the next five years [10]. The prospective economic gains that can be obtained from IoV is roughly calculated to be around 210–740 billion per annum in the next five years [19].

### 1.3. Related Work

Various reviews and surveys have been completed in IoV and ITS. Cheng et al. [20] has performed a survey on routing algorithms in IoV. They consider topology, position, map, and path routing for their survey. They recommend that the researchers should test their algorithms in scenarios, such as large-scale heterogeneous networks along with small-scale homogeneous systems to enable IoV in the real-world scenario. Tuyisenge et al. [21] have conducted a survey based on network architecture in IoV. They study some protocols and provide some information related to the mechanism of IoV in various other networks. Hussain et al. [22] have completed a review of QoS issues in IoV. In this review, they have concluded that optimal solution using QoS parameters can be used for development of IoV solutions. Ji et al. [23] have explored literature review including basic VANET technology, various network architectures, and  applications of IoV. They also proposed a design of a vehicle–road–cloud collaborative integrated network with greater throughput, lower latency, higher scalability and security. Mollah et al. [24] have performed a survey on application of blockchain for ITS using IoV in which they have studied various key challenges where blockchain is applied in IoV. Xu et al. [25] have conducted a survey on applications of artificial intelligence (AI) for edge service optimization in IoV. They study the edge service frameworks for IoV and explore the use of AI in server placement and offloading of services. Kayarga and Kumar [26] have reviewed on the bio-inspired algorithms in IoV applications, which are used between vehicles, humans, and things. Ksouri et al. [27] have conducted a survey of routing protocols with an insight into the design of geographical protocols. They have also studied various optimization techniques and paradigms for efficient routing.

There exist various optimization algorithms for network management in IoV to optimize deterministic problems. However, those algorithms are unable to tackle the probabilistic counterpart, i.e, the randomness involved in the traffic systems. Hence, to handle such random scenarios efficiently, there is a need for a better bio-inspired optimization algorithm. This survey mostly focuses on network optimization using bio-inspired algorithms. It also highlights some of the simulation tools used in the IoV.

### 1.4. Paper Organization

The remaining contents in this paper are arranged as mentioned. Section 2 provides a background of IoT and and its role in ITS. Section 3 explains the movement of VANET towards IoV and architecture of IoV. Section 4 discusses different optimization techniques for ITS available in the literature. Section 5 discusses modelling environment and steps involved in simulation. Finally, Section 6 summarizes the conclusions and future scope.

## 2. Background of IoT

### 2.1. IoT Evolution

Kevin coined the term IoT in 1982 [28], in the US, when a connection between a cola vending machine and the Internet was established to inspect the amount of cola in the machine [1]. The IoT network is a collection of interconnected physical objects, including a data processing device, people, mechanical and digital machines embedded with software and electronic circuitry which enables these objects provided with unique identifiers to collect data and exchange accordingly. IoT devices will be more than seven times the present world population, as per a report [29]. Cisco expected the number of IoT connected devices to surpass between 50 billion in 2020. The IoT is accompanying the following advantages:Allows connectivity between devices to develop smarter territories;Making one’s life easier and comfortable through allowing automation;Allows organizations to maximize efficiency and bring down costs;Allows firms to deal with wastage and improve the deliverance of services;Enables firms to develop and merge business models and improve productivity.

Along with this, new technologies and mechanisms have come up and have been advancing, such as wireless and sensor technologies, machine-to-machine (M2M) communication, big data analytics, artificial intelligence, and machine learning. With the increased number of gadgets on the network, the connectivity of heterogeneous gadgets imposes numerous new challenges. Such technologies and associated frameworks have given way to many extremely attractive IoT applications [30,31,32,33].

### 2.2. Difference among M2M, IoT, and IoE

The Internet of Everything (IoE) emerges as an advancement of IoT and it encompasses IoT and IoV. IoE is a connection of data, process, people, and things that are connected to change information into actions for creating more opportunities and better experiences. It is natural that all gadgets will be linked to the Internet in the future and all undertakings will be linked by device to device (D2D) correspondence. The accompanying advancements are assuming an indispensable part of the IoE. It joins communication from M2M, Machine to People (M2P), D2D, and People to People (P2P) [34]. The IoE is used in various applications with great execution and reaction time. Specific uses of IoE in ITS include self-driving cars, smart parking systems, smart traffic monitoring, connected cars, smart cities, and wearables (health monitoring of patients). Different smart wearable gadgets are distinguished to gather different health statuses, pulse, body glucose level, internal heat level, physical movement, etc. [35]. M2M is a subset of IoT that exist without the Internet which empowers ubiquitous networks among gadgets. The IoT, has evolved on the basis of M2M, that aims to offer many more functionalities, such as enabling communication among the same kind of machines, uniting distinct devices and systems to use different technologies, and provide interactive and fully-connected networks across varying environments. Some of the utilization of M2M communication are home and office security frameworks, traffic light frameworks, robotics, sensor networks in a petroleum processing plant, and so on. The previously mentioned uses can move the data to the server or client with good reaction time. To distinguish and accumulate diverse data from IoT gadgets, there is a requirement for cutting edge high-speed wireless network innovations, for example, 4G to 5G Networks. This advancement will have the option to satisfy their cases of high throughput, super-low latency, high reliability, accessibility, and transferring data to the client and server. This is additionally used to conquer the different issues in IoT, IoE, and M2M, such as network connectivity, and increase the speed of data flow between the numerous IoT gadgets. The above cutting edge innovations, for example, IoT, IoE, M2M, and 5G network convergence, are used to improve our lives.

### 2.3. IoT for Connected Vehicles

IoT is directly clearing a path for an associated future where vehicular nodes, gadgets, and other individual elements speak with one another in consolidated frameworks. ITS frameworks are a coordinated gathering of advancements, for example, IoT, IoV, 4G-LTE, 5G, RFID, and GPS. To manage the developing traffic requests in current urban areas, ITS frameworks, in general, adjust frameworks with a decentralized design. Vehicle information is accumulated utilizing traffic cameras, onboard sensors, radio frequency identification, infrared sensors, and information is transferred to the smart transport management system via Wi-Fi to automate and coordinate traffic signals and traffic monitoring. A service-centric heterogeneous vehicular network modelling for connected traffic environments is proposed in [36]. Safety and administration related applications in this area are sorted into three kinds.
Safety applications;Efficient traffic management;Support and infotainment applications.

#### 2.3.1. Safety Applications

The foremost target of security and safety applications of vehicular networks is to maintain a safe distance from road mishaps as it involves lives. These applications are vulnerable against delay, security applications are required to work proactively to advance the driver and, as such, ultimately prevent the disaster from happening. On the chance that a mishap has happened, this application intends to give emergency vehicles at the earliest. A new architecture has been suggested to prevent intersection collisions build on DSRC [37]. It focuses on establishing secure RSU communications deployed near the intersection area where nodes exchange their status updates. Warning for traffic signal violation, i.e., at the traffic light signal, if the driver does not stop. Notifications can be obtained when RSUs relay traffic light signals while positioning RSUs with a traffic light controller [38]. A cooperative driving of automated vehicles using B-splines for trajectory planning has been proposed by Van et al. [39]. Logical scenarios parameterization for automated vehicle safety assessment in cut-in scenarios from Japanese and German highways has been surveyed in [40]. A framework for vehicle dynamics model validation has been proposed by Widner et al. [41]. Cao et al. has proposed an improved motion control with cyber-physical uncertainty tolerance for distributed drive electric vehicle [42]. Some of the other solutions are reported in the literature—mobile crowd sensing for traffic prediction in IoV [43], and hybrid recommendation system architecture for early safety predication using IoV [44].

#### 2.3.2. Efficient Traffic Management

Intelligent traffic applications help by improving the progression of traffic and keeping away from the street clog. Vehicular nodes are informed about traffic situations early based on communication received. This may assist vehicles with changing their courses in case of traffic congestion and minimizes travel time [45,46]. The application for road congestion control helps ensure free flow of traffic by reducing road congestion. In addition, this increases the flexibility of the road and prevents traffic jams [47]. As per the necessity, before heading to a new area for direction, drivers can download maps of areas. It would also maximise the traffic flow instead of becoming trapped on the wrong road. The portal for accessing the content map database enables access to useful knowledge from home stations or mobile hot spots [48]. Non-signalized intersection network management with connected and automated vehicles is also proposed in [49]. Applications on intelligence, surveillance, and reconnaissance missions in cooperative routing problem for ground vehicle and unmanned aerial vehicle has been suggested in [50]. Niu et al. has presented an in-depth survey of space–air–ground integrated vehicular network for connected and automated vehicles and presented challenges and solutions [51]. A cost-effective traffic signal control was proposed in [52].

#### 2.3.3. Support and Infotainment Applications

Support and Infotainment utilization are intended to upgrade the client’s comfort. Likewise, health monitoring applications give utilities to patients in crises. Gaming, file sharing, searching for the nearest milk parlour, theatre, cafe, open parking room, internet video streaming, carpooling, and network service provisioning are some of the examples of infotainment technologies. As it increases network reliability and available bandwidth with an increase in peer capacity, the peer-to-peer file sharing technology has benefits over the client-server architecture. One of the frequently used P2P applications is Bit Torrent. There is a recommended emergency routing protocol called VehiHealth to provide patients with pre-medical care by providing quick communication between hospital and ambulance [53,54,55]. Computationally efficient non-linear one- and two-track models for multi-trailer road vehicles is proposed in [56]. Xin et al. [57] have proposed an AI-based QoS optimization for multimedia transmission in IoV. They present a system for multi-modal communication in which multimedia IoV transmission through mobile devices offers the quality of experience optimization model. Musa et al. [58] have proposed a design of an information-centric network with mobility-aware proactive caching scheme to provide delay-sensitive services on IoV networks. The applications of IoT for connected vehicles are listed in Table 1.

## 3. Towards IoV

Wireless ad hoc networks is a class comprising wireless networks, such as (i) mobile ad hoc networks (MANET), (ii) vehicular ad hoc networks (VANET), and (iii) wireless sensor networks (WSN) [59]. Generally, the WSN is classified into infrastructure and infrastructure-less networks. Based upon the geography and arrangement, ad hoc systems might be classified as homogeneous and heterogeneous systems. A homogeneous network is formed from similar nodes whereas a heterogeneous network is formed from dissimilar nodes. The concept of ad hoc networks is old which began in 1972, i.e., DARPA packet radio network, ALOHA, PRNET, etc. [60].

Infrastructure-less MANET is an organization of mobile devices, connected over a wireless network and follow different properties, i.e., self configuring, self healing, self protecting [60]. In MANET, because of mobility, frequent link breaks, and dynamic topology, nodes in these networks act as routers to transfer packets. It enables spatial spectrum reuse due to the limited bandwidth of each node, another type of ad hoc network is VANET which is shaped by various vehicles present on the road.

Different vehicles communicate with one another on the road and each of these vehicles has a tool called an On Board Unit (OBU). OBU can talk to vehicles and RSUs, which act as access points [61]. Vehicles are enabled to talk to one another in different ways. VANET is very helpful in spontaneous data exchange and it is a key component in ITS. It offers different applications concerning highway traffic, road congestion and accidents. The two communications considered in VANET are vehicle-to-vehicle communication (V2V) and vehicle-to-infrastructure communication (V2I)—between vehicles and roadside access points. VANET looks very similar to MANET, but it is slightly different in following route patterns, they follow predictable mobile patterns whereas MANET has unpredictable mobile patterns. The architecture of VANET is shown in Figure 1.

There are extensively three distinct segments of VANET, one is an OBU which is answerable for information collection from various sensors and other vehicles. The second-RSUs which offer an infrastructure that enables communication to the external network. The third communication technology helps these units to talk to each other, IEEE 1609.2 is commonly known as Dedicated Short Range Communication (DSRC) 802.11p [62]; vehicles must be equipped with the IEEE 802.11p based OBU and DSRC, along with added sensors to be completely aware of the condition where the vehicle find itself in [45]. A cooperative perception technology of autonomous driving in the internet of vehicles environment survey is reported in [63]. The VANET architecture is commonly divided into three sorts of classes namely:Mobile and wireless LAN networks which are used to direct and obtain traffic data through fixed portals and WiMAX/Wi-Fi;Pure ad hoc, that is, between vehicular nodes and defined gateways;Hybrid, that is, blend of infrastructure and ad hoc networks.

Different standard structures of vehicular networks consolidate CALM (Continuous Air Interface for Long to Medium range) by ISO [64], and C2CNet (Car-to-Car Network) by C2C Consortium [65], and WAVE (Wireless Access in Vehicular Environment) by IEEE [66]. The IoV is viewed to be a development pertaining to V2V network [4]. The IoT is supporting the demand of traditional VANET to IoV scope, the IoV coordinates vehicular networks with data storage and data analytics services. The IoV brought intelligence to the communication of the vehicular network, which improves driving support for completely autonomous driving by facilitating the AI awareness to the encompassing vehicular situation. The components of IoV include [4] vehicles (network formed by vehicular nodes), RSU, infrastructure (street and traffic-related sensors), personal devices (smartphones and PDAs), and people (drivers). Accordingly, the different services that are possible are listed as follows.
Vehicle-to-vehicle information services;Infrastructure and vehicle information services;Sensors and vehicle information services;RSU and vehicle information services;Human and vehicle information services;Vehicle and personal devices information services.

### 3.1. Architecture of the IoV

Numerous models have been proposed by researchers for IoV [4,67]. These architectures are helpful to meld various kinds of interactions. The basic IoV innovation stack incorporates three layers, namely: perception, networking, and the Internet and service platforms. The key concept of IoV is specified by its three-layer architecture. In the literature, we can find two more layered architectures—the five-layer architecture [68] and the seven-layer architecture [69], which also include the processing and business layers.

#### 3.1.1. Layer 1: Perception

Sensors are implanted in the physical environment to collect and transfer data. Sensing devices do not associate legitimately to the web, they can synchronize with phones and different gadgets utilizing Bluetooth LE. Raw information was examined against the characteristics of the global positioning framework obtained through the sensors. IoT sensors are a fundamental aspect of IoT innovation.

#### 3.1.2. Layer 2: Networking

Micro controllers regulate the information channeling to IoT sensors and various actuators. Micro controllers and Internet connectivity, share information obtained at the first layer and examine for second layer to make a further move. Networking, either wireless or wired, is the basic and most important responsibility of this layer.

#### 3.1.3. Layer 3: Application

Service platforms take measures to adjust, alter, maintain, and monitor physical conditions after data investigation. Telematics, data mining, voice over Internet protocol (VOIP), blockchain, and Cloud SaaS stages are typical utilities.

The following data were taken from scientific publications related to attribute measuring using IoT sensors, methods of information transmission between smart sensors and actuators, networks, and protocols used in communication methods. Approaches to data storage can be seen in Figure 2.

### 3.2. Network Protocols Used in Vehicular Networks

IoV systems include a network of vehicular nodes, fixed RSU, and a central server. Road-related alerts relevant to peer drivers are interchanged in V2V communication [70]. On the other hand, V2I communication is utilized for the collection of sensor information and dissipation of alerts based on locations to vehicles [9]. Mobile networks, DSRC/WAVE, Wi-Fi, and ZigBee are among the innovations for the wireless channels. DSRC and OBU combined with extra sensors are necessary to be completely conscious of the circumstance in which the vehicle finds itself in [71,72,73,74]. The IEEE 802.11ah long-range Wi-Fi, which can be supportively used in vehicle frameworks, especially when vehicles are spread over 1 km in road organisations. During conditions where the vehicles run at 160 kmph, the utilization of WiMAX might fit better [75]. The four driving radio access technologies (RATs) for V2I communication are 4G/long-term evolution advanced (LTE-A), 5G, Wi-Fi, and DSRC. As of now, car companies are pursuing numerous for open RATs to empower applications both for protection and security (primarily web access). Audi and Volvo have a lot number of vehicles that have Internet accessibility regulated by 3G, 4G LTE, and 5G [76]. Shah et al. [77] have proposed a novel cluster-based MAC protocol (CB-MAC) for VANETs and they have optimized the CB-MAC protocol [78]. Karabulut et al. [79] have proposed a multiple-input multiple-output (MIMO) and orthogonal frequency division multiplexing (OFDM) based MAC protocol which uses the advantages of both the techniques. Wu et al. [80] have designed a self-adaptive time division multiple access (TDMA)-based MAC protocol for VANETs to improve the stability of the time slot scheduling in VANETs. Han et al. [81] have proposed an adaptive time slot access MAC protocol in distributed VANET which improves the time slot access efficiency by adapting the access time slot according to the driving direction of the vehicle and the traffic density ratio.

The various communication technologies applicable are shown in Figure 3.

### 3.3. Routing in IoV

Many real-life situations need different technologies for vehicle networking. For instance, driving in metropolitan circumstances, drivers know the traffic conditions of urban roads ahead and alter their route direction according to the traffic situation on their route. By utilizing the cutting edge innovation of IoV, individuals can likewise diminish fuel utilization and environmental pollution. A significant exploration aspect in IoV is its routing conventions. Researchers have proposed different routing algorithms for IoV characteristics, i.e., heterogeneous communication range, dynamic topology, geographically constrained topology, mobility of the vehicles, time-dependent vehicular density, ad hoc network, and the elements that construct the network are vehicles. In recent days, most routing protocols on varying factors, such as energy, network lifetime, efficiency, scalability, multi casting, reliability, and load balancing have been made to satisfy IoV prerequisites.

Depending upon the number of senders and recipients participating, routing approaches can be sorted into three kinds: geocast/broadcast, multicast, and unicast approaches. Second, we arrange them into four classifications dependent on data needed to carry-out routing, i.e., map, topology, position, and path-based. Third, it is grouped to be delay-sensitive and delay-tolerant. Lastly, we identify protocols based on applicability in their various dimensions, for example, 1-dimensional, 2-dimensional, and 3-dimensional. The target networks we talk about are heterogeneous and homogeneous. The routing conventions that are often utilized (traditional routing algorithms) are dynamic source routing, optimized link state routing, ad hoc on-demand distance vector, geographic source routing, and greedy topology. These routing conventions are utilized to transfer the data packets between vehicular nodes. Under these routing conventions, the proactive approach relies on routing strategies associated with a table-driven method. Proactive routing conventions generally rely upon algorithms related to the optimal route. They store all the gathered information identified with the associated vehicular nodes in related predefined tables, as well as being the primary component of routing conventions. Each table in this approach is refreshed by its vehicular node when the network topology changes. Reactive routing conventions depend upon algorithms identified with on-demand actions. At the point when two vehicle nodes need to interact, they begin the path discovery of the route and one of its fundamental advantages is the reduction in network traffic [82]. Geographic dependent routing conventions are dependent on situations corresponding to the position technique utilizing area-based applications, such as GPS. Geographic applications are used while giving information for path selection [82]. In different scenarios, these traditional routing algorithms performed comparatively better, but failed to provide the optimal routing solution in IoV environment. To enhance the reliability of safety applications, bio-inspired approaches are used. Similarly, recently proposed protocols based on SI techniques, namely, ant colony and particle swarm algorithms, etc. In comparison with traditional techniques, these protocols have performed better individually.

## 4. Network Optimization

IoV, the evolving type of VANETs and MANETs, is more exceptional but more complex to adapt. Special IoV capabilities include high-processing, high-speed data access, robust usability, and variable network density. It is a hard job to prepare an efficient optimal routing protocol for information transmission in IoV, to hold all diverse aspects of vehicles on the route. Heterogeneous node density and networking, inconsistent connectivity, and variable mobility must be taken into account in an optimal routing protocol. Optimization is characterized and portrayed as the innovation utilized to enhance the performance of the network for any circumstance. Optimization is a numerical issue experienced in all engineering disciplines. It implies finding the most ideal/desirable solution. Optimization issues have widely occurred and, hence, various techniques for taking care of these issues should be an active research topic. Optimization algorithms can be either stochastic or deterministic. Strategies to tackle optimization issues require a lot of computational power, which will generally tend to fail as the problem size increases.

### 4.1. Optimization Techniques

Optimization generally parts into two different classes, one is deterministic algorithms and the other is stochastic algorithms. The issues that cannot be settled from the deterministic algorithm are named as non-deterministic algorithms and, thus, the novel solutions developed by stochastic algorithm assists with taking care of the non-deterministic problems up to an optimum standard. The non-deterministic problems are solved by meta-heuristic algorithms. Meta-heuristics are characterized as heuristics at a more significant level of frameworks. Thus, meta-heuristics are problem-independent approaches, though heuristics are problem specific. This heuristic technique helps with taking care of the complex problem [83]. Biological behaviour-influenced algorithms are known as bio-inspired stochastic algorithms. Bio-inspired stochastic algorithms have gained importance, especially for tackling complex enhancement issues of routing. These stochastic procedures that are created to accomplish near optimal solutions for huge scope optimization problems [84]. The traditional solution of NP-hard problems with a numerous variables and non-linear objective functions will sometimes fail (being stuck in a local optimum), leading to the development of alternative solutions. These techniques are moved by the characteristic biological evolution, as well as the social conduct of species. These approaches rely on natural patterns and behaviours that have the capacity for self-adaption and self-association and are utilized as powerful optimization tools.

Apart from these, there are other optimization methods in which the researchers have focused on vehicular congestions in urban areas. Authors proposed a centralized simulated annealing method for alleviating vehicular congestion in smart cities [85]. They used a novel dynamic centralized simulated annealing-based approach for finding optimal vehicle routes using a different type of cost function. Amer et al. proposed a hybrid game approach-based channel congestion control for IoV [86]. They developed a new hybrid game transmission rate and power channel congestion control approach on IoV networks, where the nodes play as greedy opponents demanding high information rates with the maximum power level. A new congestion control approach is proposed which is based on the concept of hybrid power control and contention window to ensure a reliable and safe communications architecture in IoV [87]. Fu et al. proposed an IoV system assisted by mobile edge computing which is used for cross-layer offloading to provide low latency and abundant computation resources [88].

Bio-inspired optimization techniques, swarm intelligence (SI), evolutionary algorithms (EAs) play a crucial part in the computer intelligence sector that has become famous over the recent times [89,90]. EAs which depend on Darwin’s hypothesis of natural selection and survival of the fittest. EAs and SI are taken from biological evolution processes, which depend on behavioural models of social animals, for example, ants, bumblebees, fireflies, fish, flying creatures, etc., hence, they look for food substance or a better environment, EAs commence with a collection of candidate solutions, create alternatives for offspring recursively, and evaluate solutions till an acceptable solution is found. Genetic algorithms (GAs) [91], evolution strategy (ES) [92], evolutionary programming (EP) [93], genetic programming (GP) [94], estimation of distribution algorithms (EDA) [95], differential evolution (DE) [96] are popular algorithms in this EA group. SI algorithms begin with a group of possible candidate solutions, and in every iteration, a novel group of candidate solutions is produced derived from verifiable and other applicable historical data. A few models of this kind comprises of an ant colony algorithm (ACO) [97], particle swarm optimization (PSO) [98], artificial bee colony optimization (ABC) [99], firefly algorithm optimization (FA) [99], salp algorithm (SA) [100], bacterial foraging optimization (BFO), artificial fish swarm optimization (AFS), etc. The bio-inspired optimization techniques are shown in Figure 4.

### 4.2. Evolutionary and Bio-Inspired Algorithms

The SI algorithm understands the solution for the problem by learning from certain life or natural phenomena [101]. These types of techniques consolidate the self-association, self-learning, and self-versatile characteristics of the regular nature. In the computation cycle, the general population is searched for the solution space through the acquired estimated data. In the course of a search cycle, the populace advances by the fitness function values, which are fixed beforehand. Consequently, the algorithm has certain intelligence. Inferable from its points of advantage, when the SI algorithm is utilized to resolve a problem, it is not important to manage the solution issue ahead of time to acquire a detailed solution. It is, hence, conceivable to effectively tackle some highly complex problems. The swarm intelligence has proved its efficiency in tackling the routing issues in such self-organized systems as MANET, VANET, WSN, and IoV. ACO and PSO are the traditional SI optimization techniques. ABC, BFO, FA, AFS, and several others are less well-known techniques. Initially, Swarm techniques were intended for stationary optimization problems. Among them are ACO, ABC and PSO. Bees, ants, salps, and other swarm activity that resembles that of the nodes in the wireless ad hoc network. The most widely used SI algorithms listed in Table 2 and are discussed below.

#### 4.2.1. Genetic Algorithms

GA portrays biological advancement as a problem-solving approach. GA operates on the search space, i.e., population or populace [122]. Every component in the population is labeled as a chromosome. GA begins arbitrarily with arbitrarily choosing a group of suitable solutions from the populace. Every chromosome is an answer of its own, the chromosome is assessed for fitness and this characterizes the solution. This algorithm utilizes an adaptive heuristic inquiry strategy that searches the set of the finest solutions from the populace. New off-springs are created and advanced from the chromosomes utilizing operators, such as selection, crossover, and mutation. The fittest chromosomes are transferred to the upcoming peer group. Weak chromosomes have a lesser possibility of moving to the future generation. This is because GA depends on Darwin’s theory of evolution that expresses that “survival of the fittest”. This procedure repeats up to that point at which the chromosomes have the best fit for the given problem. The outline implies that the mean fitness of the populace increments at every cycle, and by iterating better outcomes are found. GA gives alternative strategies to tackle problems that are hard to unravel utilizing conventional techniques. For instance, in [123], GA has been suggested to take care of the issue of optimal deployment of WSN for increasing probability of searching an moving object in the field. In [124], the authors utilized GA to tackle the multi-objective optimization formulation utilized to attain the ideal stationing of sensors at the point of port entry to inspect the vessels and identify the movement of illicit freight. To unravel an optimization problem with multiple targets, a GA-based normal boundary intersection algorithm was utilized [125]. The issue involved advancing the sensor field setup for the discovery of the target in motion. An advancement procedure with multiple objectives has been suggested in [126] for the task scheduling for WSN. GA-PSO (particle swarm optimization) is a hybrid algorithm consolidating GA with PSO [103]. PSO offsets the limitations of classical GA [104] and presents an order-aware hybrid genetic algorithm (OHGA), consolidates two heuristics techniques to address capacitated vehicle routing problem (CVRP), which targets setting a minimal cost course path. GA is used to unravel and develop Universiti Tenaga Nasional (UNITEN) bus routing to restrict the time taken for all stops to cover the entire distance and minimize transport costs, contributing to the rapid transport of students to their locations [105]. The intelligent signal light scheduling is upgraded by many specialists using the real-time traffic flow. The GA-based step re-allocation algorithm can deliver good execution while retaining real-time execution, outperforms conventional methods.

#### 4.2.2. Ant Colony Optimization

ACO mimics the searching conduct of actual ants. The objective of the ants is to locate the shortest path from their home to their sustenance sources [127]. As they continue looking for food sources, ants glance around arbitrarily. At the point when they discover one, they return to the home, setting out a fragrant stuff on the ground, called a pheromone. The measure of pheromone is identified with the quality of the food source, as determined by the amount of food and how far it is away. The other ants will search less haphazardly, since they shall be pulled in by the pheromone trails. Numerous ants will be pulled on tracks with more pheromone, which will, thus, prompt increasingly more pheromone to be set in these ways. At last, all the ants will be drawn into the best route. The indirect communication plot through pheromone trails prompts this optimization, this phenomenon is known as stigmergy. Diverse analogies based on ants are stated in the literature, beginning from the original ant system (AS) [128] to later variations, similar to the ACO [129]. Although insect-based frameworks have initially been tried on the travelling salesman problem (TSP), numerous other combinatorial issues have been addressed since then. In the following, the essential AS algorithm used for solving the TSP is expressed and a portion of its variations are quickly presented [128]. Applications for vehicle routing issues are reviewed. Dynamic route optimization which uses an algorithm that is inspired by nature is constructed by [130]. The notable algorithms that are based on nature are PSO and ACO for route planning in IoV, where ACO gave good results over PSO by giving the short distance routes which need less travel time. This research [107] centers predominantly around dynamic communication range per each vehicle in IoV organization. Here, clustering-based ant colony optimization is proposed (CACOIOV) and dynamic aware transmission range on local traffic density (DA-TRLD), is utilized along with CACOIOV to give a routing model, to upgrade route discovery and maintain network stability in IoV network. A recent ant colony optimization (ACO) algorithm [108] offers an agent-based paradigm designed for inbound logistics to solve a capacitated vehicle routing problem (CVRP). The model was experimented utilizing input information supplied by Gali Group, a logistics company in Sicilia region of southern Italy. Through better routing, fewer kilometres covered and load factor, for the logistics company to increase revenue. This article [109] proposes a maiden look-forward preventing heavy traffic creation and accidents, based on IoV traffic management. The suggested practice is seen via segmenting road maps into a small number of maps. To determine the optimal path, the ant colony algorithm is used to every small map. In turn, in this study, the Fuzzy logic-based traffic intensity measurement feature is suggested to model heavy traffic congestion.

#### 4.2.3. Particle Swarm Optimization

PSO mimics the conduct of fish tutoring or flocking birds. Eberhart and Kennedy [110] introduced PSO for addressing continuous optimization issues. Each bird or particle in swarm speaks a possible answer to the issue. To be more precise, every particle comprises of a velocity and position vectors, these are refreshed in accordance with the best position of a particle and swarm. PSO algorithm has two basic modelling techniques, for instance, the global best and neighbourhood best. In the global best, the neighbourhood comprises the particles in the entire swarm, which move and offer data. In the neighbourhood best, the neighbourhood of a particle is determined by some fixed particles. Poli et. al [131] expressed that the global best model converges quicker than the neighbourhood best model. The previous model is more likely to become trapped in a neighbourhood optimum than the latter model. PSO varieties can be found in [132,133]. The global best model has good impact in multi-swarm [134,135,136], while for algorithms with a single swarm, the local best model is mostly used [137,138,139]. One of the best answers for network adaptability in IoV is a clustering-dependent model. Vijayalakshmi and Anandan [111] suggested a PSO and Tabu hybrid algorithm called Tabu-PSO to pick the CH with the lowest energy consumption in the cluster, to increase the capability to choose CH in IoV network. A PSO-based routing algorithm was proposed by Wang et al. [112], which merges virtual cluster and mobile reception technology. The algorithm considers residual energy and node location for selecting CH compared to Tabu-PSO. Hasan and Al-Turjman put forth a version a kind of bionic PSO fault-tolerant routing algorithm by expanding upon the current method, multi-objective optimization, for rapid recovery from the path failure [113].

#### 4.2.4. Artificial Bee Colony Optimization

ABC algorithm mirrors the conduct of bees’ colonies [140]. A customary ABC algorithm comprises food sources, while every food source represents a possible fix for the issue. Food sources are upgraded by bees gathering, i.e., employed, onlooker, and scout bees. Every honey bee in the province delivers another candidate food source position out of its former position. If better food sources are discovered, the new courses of action have a better fitness over the present, the better ones are revived. The relative probabilities from fitness chosen from the employed bee stage are settled in the onlooker honey bee stage. Then, onlooker bees choose an answer in which the best solution possesses a better chance of probability to be picked by onlooker honey bees.

From that point forward, onlooker honey bees act similar as the employed honey bees do. Eventually, scout honey bees haphazardly reassign solutions, that they are left behind as if they have not been upgraded for a specific duration. There are a few progressions of honey bee techniques, for example, ABC, bee colony optimization, virtual bee algorithm, honey bee mating optimization, and beehive algorithm. Surveys on the various advancements can be obtained from [8,141,142]. The ABC algorithms which have earned most of the consideration, particularly in discrete optimization problems [142]. Just like the ACO, ABC algorithms are highly flexible to deal with discrete optimization problems. Combinatorial optimization, for example, routing and optimal paths have been effectively resolved by the Honey bee and ACO. While they can solve both continuous-discrete optimization problems (DOP). However, it must be noted that they should not be the first preference for continuous problems. Garg et al. [114] ABC anomaly detection with a Cauchy-based mutation operator consists of several stages: (a) collection of suitable feature set, (b) optimization of Support Vector Machine (SVM) parameters, and (c) arrangement of vehicular traffic. Cauchy-based ABC strengthens the optimizer’s local search capacity with faster convergence. The final step of classification of data is then carried out with a refined set of parameters using SVM. Alzaqebah et al. [115] introduces an algorithm for the vehicle routing problem with time windows (VRPTW). To increase the solution efficiency of the original ABC, an updated ABC Algorithm is suggested. The high exploration potential ABC slows down its speed of convergence, which may replace abandoned (unimproved) solutions with new ones because of the mechanism used by scout bees. Masutti and Castro [116] TSPoptBees provides a better approach to the most notable problem of vehicle routing: TSP. To solve continuous optimization tasks, TSPoptBees, another technique (optBees), was introduced and therefore built to resolve this class of discrete optimization functions.

#### 4.2.5. Firefly Optimization

This algorithm depends on the flashing conduct of fireflies. Contrast to ACO, where different ants are pulled in by pheromones, fireflies utilize a flash signal system to attract other flies. It was motivated by the flash patterns and firefly’s conduct [140,143]. A FA depends on three assumptions:Every firefly can be pulled in by other fireflies;Every firefly’s appeal is proportional to how bright the other fireflies are;The problem scene determines the quality of fireflies.

In this manner, less bright firefly will advance towards a brighter one. If a firefly cannot find a brighter firefly, it will move haphazardly. Every firefly shines relatively to its solution quality, which, along with its appeal, directs how strongly it draws in different individuals from the swarm. Some pre-imperative assumptions in this algorithm are that the fireflies are uni-sexual, the appeal is proportional to how bright they are, and a firefly will move haphazardly if there is no firefly having more prominent splendour. The current FA has pulled in much consideration [143,144]. The NP-hard arrangement problems [144] can be adequately addressed by a distinct form of FA, whereas a bare basic investigation has shown the viability of FA over a extensive scope of test problems, comprising multi-objective load dispatch issues [143,145]. The multi-objective approach to the firefly algorithm (FA-OLSR) [118] was simulated and the outcomes of the simulation divulged and enhanced the ratio of packet transmission, mean routing load, and end-to-end latency. A new variant named FA with neighbourhood attraction (NaFA) was introduced. According to Wang et al. in NaFA, each firefly is lured by other more bright fireflies chosen from a defined neighbourhood which is already determined instead of those from the whole population [119]. Using many well-known benchmark features, experiments are performed and the results show that the suggested strategy can increase the accuracy of solutions efficiently and reduce the complexity of computational time. Dhanare et al. proposed a hybrid optimization approach that combines modified ant colony and firefly optimization techniques (MAF) to calculate the average speed and find the best route to the destination. The MAF algorithm combined attractiveness and pheromones to find the optimal path and reduce the travelling time [146].

#### 4.2.6. Salp Swarm Optimization

A recent meta heuristic, salp algorithm (SA) is influenced by the swarming behaviour of slaps in the oceans. It is an optimization method based on population, put forward by [100]. By the following statement, the SA’s actions can be convinced, i.e., the objective of this swarm is a food source in the search space, and the salp chain helps to search for optimal food sources. In SA, the end-to-end individuals in the salp swarm are split into two groups to model the salp chain formed by individuals. The individuals in the swarm are classified into two parts: leaders or followers. The salp chain starts with a leader who decides the direction of travel and forages the population’s route and directs the salps chain toward the food and the rest go after a leader to guide them to establish a chain structure. The followers will arrive at a position which is better as compared to the present best solution (food) in the procedure of going after the leader to upgrade the position. The food is placed to the better position at this stage, and the updated leader directs the followers towards food. The aim of the optimization problem is to determine the global optimal value, so the global optimal value is utilized as the food that needs to be identified by the salp chain. The position of the global optimal value in the optimization problem is not known. The salps Chain model can be pushed closer to the target value by considering the optimal value in the current iteration as the global optimal value. The entire salps chain can be taken closer to the food chain, according to the leader’s position of the food update. The SA algorithm is used in this paper [147] for route planning, which is an NP-hard optimization problem. Its outcome is compared with deterministic and other nature-inspired algorithms. The findings show that SA is better than all the other meta-heuristic algorithms in route planning. This approach improves the average cost and total time taken when compared to other algorithms. Route planning is utilised in several real-life instances, such as self-driving car, robot navigation, autonomous UAV for search and rescue operations in dangerous ground-zero situations, surveillance of civilians, military combat, and commercial services, such as package delivery through drones. To optimize coverage and the radio energy model to minimize consumption of energy, the paper [148] proposes a weighted distance location update called weighted salp swarm algorithm (WSSA). The optimal problem with sensor deployment is known to be a multi-objective problem. The majority of the previous research work is based primarily on solving only a single objective of the problem. This paper aims to address both the coverage and energy issue at the same time. The WSSA algorithm has been found to outperform all the other stochastic algorithms in maximising coverage and energy efficiency of WSN.

## 5. Modelling Environment

Simulation modelling plays an important part in scientific research. Researchers often apply it in the design tool to understand the protocol’s behaviour and to evaluate the network productivity. One of the most crucial tasks is to identify a suitable network simulator. Many network simulators are available for research. They are providing platforms for testing, modifying, and evaluating protocols in IoV. The following combination of simulation tools were utilized for better performance, such as Open Street Maps (OSM), Objective Modular Network test bed in C++ (OMNeT++) version 6.0, Simulation of Urban MObility (SUMO), Vehicles in Network Simulation (VEINS) version 5.2, Network Simulator (NS) version 3, and MATLAB. All the systems mentioned above are used for modelling IoV Networks and allow:Enhancing efficiency in implementation;Experimenting real network deployment in simulators;Conducting scientific examinations in this field;Reducing implementation and deployment cost of real network.

Integrated framework is provided by IoV simulators to provide simulation execution without running another software and therefore inter-dependencies are solved. Categorization of simulators can be performed in three ways: (i) mobility generators, (ii) network simulators, and (iii) integrated software for mobility generators and network simulators.

OSM, an open source tool [149], where clients can alter the map data. Geo-information assembled by the clients shall be viewed as the essential project. Since 2004, the quantity of OSM clients has expanded more than 2M. Each enlisted client can alter the OSM data. The OSM sent information is utilized in creating paper maps, electronic maps, route planning, and geo-coding. While it cannot be considered as a simulator, but it can be seen as a tool that supplements simulations. To use real-world maps, the client can import data into simulators, while importing geo-data from a free-source database. If simulations use real maps, hours spent on manual creation of a map can be lowered, allowing more practical simulation.

Objective modular network testbed C++ (OMNeT++) [150], C++ discrete event simulator based on open source component with GUI help. To form the simulation set-up, a few modules are stuck together. OMNeT++ output text files (CSV, JSON, or SQLite Scalar or Vector), can be processed with other software (MATLAB or R). OMNeT is available for scholastic and non-benefit use. OMNeT works to optimize network simulators of different types, but it really does not provide highlights of digital transformation for vehicle development.

Simulation of urban mobility (SUMO) [151] is designed primarily to model large road networks. The simulator contains settings for various road types (i.e., parking, road surfaces, highways). Using few files, setting speed limits, road network model can be built together with adjoining structures to program routes in simulation for different vehicles. It is possible to manually delegate traffic flows, calculate them based on demand data, or produce them entirely at random. The application has different plug-ins that can be used to upgrade the simulator, that works well for importing maps across the globe with OSM [152]. The traffic control interface (TraCI) [152] enables new functionality to be added to SUMO or to be linked to other software. Installing and simulating road networks is easy, but it can not simulate communication in network. To personalize visualization of vehicles, real-world road networks are imported with SUMO using TraCI to customise car visualization.

Vehicles in network simulation (VEINS) [153] is an open-access software designed to model IoV, since OMNeT++ is a simulator of network but lacks vehicle movement and SUMO is a simulator of the road network but lacks vehicular communications. VEINS is related between network and traffic simulators. For instance, on receiving an alert message created by the network simulator, vehicles need to divert itself or reduce its speed (changing the mobility pattern). Using TCP, message sharing takes place between the mobility generator and the network simulator. For road development plans, actual maps may be used. Although VEINS does not simulate the network itself, it connects two programmes, when running IoV simulation with SUMO and OMNeT++ [154].

A discrete-event network simulator (NS) [155] supports algorithms for routing, queuing, and introduction of IEEE 802.11p with respect to IoV. The simulator is applicable as it helps researchers to understand processes, along with the documentation and logs, at a deeper level for the application of protocols. NS requires multiple packages to run in connection with itself (i.e., NAM, CCC compiler, Tcl/Tk files). It is designed for the simulation of networks, which is important for IoV experiments to be carried out. NS-2 [156] is also used for simulation modelling; the C++ simulation kernel and object-oriented tool command language (OTCL) are used. It is open source; it is simple to incorporate new modules into it. It also provides wireless assistance includes node mobility, radio communication modelling, and protocol 802.11p. NS-3 [157] is the substitution for NS-2 to fulfil the advanced research criteria for the network. For simulation modelling, TCL is no longer necessary, and Python script is allowed. NS-3 does not endorse backward compatibility with NS-2. The other open-source network models have expanded device integration.

MATLAB [158] is a tool that is widely used for database evaluation and analysis. When using the built-in graphics expertise of MATLAB, simulation is usable and easy. Different circulations of data, 2-D, 3-D graphs, and animations are common investigative techniques. There’s also an extension to MATLAB-Simulink. It takes MATLAB input datasets [159] and continuously uses them to generate some output. The findings will yield to MATLAB visualization at that point. MATLAB is hierarchical, where it is easy to conduct large-scale operations and numerous people can function in it at the same period. Additionally, also available in MATLAB is versatility, as there are collections in scripting languages other than R (e.g., Python, Java, C++).

In particular, given that the software is not legitimately built as a simulator, ThingSpeak [160], an IoT analytics tool, it shows similarities in its ability to direct visualisation. It is possible to import C++, thereby allowing the inclusion of some languages. A comparison of VANET simulations is presented in [161,162]. Table 3 lists the use of various simulation tools reported in the literature.

### 5.1. Simulation Using NS3, OSM, and SUMO

OSM provides the ability to save street data in the OSM file format. This file format is OSM-specific, XML-coded, and contains organized and ordered geographical data. Through OSM, we can select our own street structure, Google maps can also be used to validate or used for naming of streets. After downloading the map from OSM, a*tcl* file is created from OSM output, which is already in the format of *osm .sumocfg*, converted into *.xml* format. SUMO has a *traceExporter.py* file, this file is processed to obtain a trace file out of the XML input file.

A *tcl* file is created with different inputs from vehicles in IoV environment. In next step, the *tcl* file is integrated with *.cc* file which consists of network parameters and protocol settings for which following *tcl* file works. The following simulations are performed for various standard routing protocols OLSR, DSDV, AODV, DSR, and analysed for different metrics, such as receive rate, packets received, MAC physical overhead, packet loss and throughput, etc. The output files generated from the above simulations are *.tcl* (metrics generation), *.tr* (ASCII trace), *.flowmon* (flow monitor) [172], *.xml* (for netanim) [173], and *.pcap* (wireshark) [174]. Based on the requirement, it is used for further analysis.

STEP 1: To Create SUMO-GUI or SUMO configuration file from OSM output.
 $ export SUMO$_$HOME=/home/harika/sumo/ $ cd sumo/tools $ python osmWebWizard.py

Once the data are generated from OSM as shown in Figure 5, move to Step 2.

STEP 2: To create mobility.tcl file (How to create?).
 $ sumo -c osm.sumocfg --fcd-output tracefile.xml $ cd $ cd sumo/tools $ python traceExporter.py -i tracefile.xml                 --ns2mobility-output=mobility.tcl

Now check the number of nodes in the *mobility.tcl* file which is very important. Move the *mobility.tcl* into the */home* folder. The vehicular nodes are shown in Figure 6.

STEP 3: Run *mobility.tcl* file with total number of nodes, duration, log file etc. The program is already in the scratch folder. The resulting simulation view is shown in Figure 7.
 $ cd ns-allinone-3.29/ns-3.29 $ ./waf --run "scratch/ns2-mobility-trace --traceFile   =/home/mobility.tcl --nodeNum=1815 --duration=100   --logFile=ns2-mob.log"

A window will be opened and select the *vehicularmobility.xml* file and run the simulation. You can do the network performance like wire shark, ASCII trace metrics using trace metrics, GNU plot for plotting the characteristics, etc. Vehicular movement, states, and packets transferred as analysed using a network animator, as shown in Figure 8.

STEP 4: Include *netanim* header file and run the simulation. Run the following command and include the line as below:  \#include "ns3/netanim-module.h"  AnimationInterface anim ("vehicularmobility.xml");  Simulator::Run()

In order to run NetAnim, the following steps have to be performed.
 $ cd $ cd ns-allinone-3.29/netanim-3.108/ $ ./NetAnim

### 5.2. Simulation Using OMNET++, VEINS, and INET

The VEINS–INET framework together helps to understand full feature set of INET framework in a VEINS simulation [153,175]. With this we can access full ipv4, ipv6 stacks, wired networking, mobile ad hoc network protocols, bit-precise PCAP traces or network emulation. So using this VEINS–INET we can also model libraries based on the INET framework. Additionally, we can use things such as simuLTE which is a 4G simulation.

STEP 1: Take map from OSM and turn it into SUMO simulation.
(i)Open www.openstreetmap.org (shown in Figure 9).(ii)Export the map by manually selecting the area, i.e., downloaded map is in *.osm* file.

STEP 2: Java Open Street Map Editor (JOSM) for editing map information. Sometimes data from OSM is not completely ready for traffic simulation; information from OSM can be enhanced using JOSM, which is a java tool for editing maps shown in Figure 10.
(i)Open *.osm* file in JOSM, dataset will be rendered.(ii)Export the map by manually selecting the area, i.e., downloaded map is in *.osm* format.

We can manually edit the properties and remove unwanted information from the map, usually lot of information will be deleted automatically when using *netconvert*. Here, we will run *netconvert* operator, we can use use specific options to run *.osm* file on highways and motorways under different speed settings.
$ netconvert --osm-file map.osm --ouput-file map.net.xml --geometry  -remove --roundabouts.guess --ramps.guess --junctions.join  --tls.guess-signals -tls.discard.simple --tls.join

These are standard options while converting *osm* file to sumo network file. The generated network file is shown in Figure 11.

After creating SUMO network based on *osm*, we use random_trips Python file with network file *.net.xml*, which is used to obtain the trips file and route file.
$ randomTrips.py -n map.net.xml -e 1000 -o map.trips.xml

After creating trips file, we need to convert them to obtain the route file by taking *.net.xml* and *.trips.xml*. After creating *trips.xml*, trips file is converted into route file, such as those shown below:$ duarouter -n map.net.xml --route-files map.trips.xml                     -o map.rou.xml

We can see the output file as *.rou.xml*, here all these files, i.e., *.net.xml*, *.trips.xml*, and *.rou.xml* are required to create SUMO configuration file. The last step is to create SUMO configuration. Along with *.net*, *.route*, and *.trips*, manual file is created in name of *map.sumo.cfg*, following XML code is added in to file.
  <configuration>  <input>  <net-file value="map.net.xml"/>  <route-files value="map.rou.xml"/>  </input>  <time>  <begin value="0"/>  <end value="1000"/>  </time>  </configuration>

We can see above simulation setup in Figure 12, map scenario and simulation of vehicular movement can be observed.

The VEINS–INET sample application is shown in Figure 13. The start application checks if the node is the first node and sets the display to be red and then sets the speed to 0 to stop it is how it actually accesses the vehicle to stop it and then that host will create a VEINS–INET sample map with the size of hundred bytes it sets the road id and then creates a packet called accident and then sends that packet along it calls the send packet which is the part of the VEINS–INET application base when it receives a packet which process packets which color the vehicle as green and then it will change the route to what ever road ID was given it is just a very simple application that shows you how you can manipulate different items in VEINS and INET. The application base extends our INET UDP socket and also INET’s application base, so this is doing all the UDP and networking for us. In order to run INET application first we need to make sure that we are running SUMO. For that we need to launch Python and SUMO application and only it will be listening to port 9999 which was set in VEINS–INET manager. The startup of the simulation, i.e., radio medium and manager is shown in the figure. We can see the vehicle transmission information in the OMNET setup.

## 6. Conclusions

Effectual communication in IoVs is troublesome task because of fast-moving vehicles on streets and the impact on data delivery conveyance. By building up our novel thoughts, we will succeed achieving our goals towards local optimum problems and staying away from network dissemination issues. The proposed algorithms in the literature show optimal performance to make proficient and qualitative vehicles to any node communications and confirm reliable data delivery to every vehicle. The significant prerequisite for a technique that is methodical and productive to modify the limits of routing is required. In network optimization, the SI algorithm is motivated by the biological phenomena in the common world, yet there is no development directing numerical hypothesis, including analysis and verification of the convergence of the algorithm. SI techniques are still at the underlying stage. This study has recorded several SI techniques. The ACO in the vehicle routing problem still has greater potential. The stochastic nature of vehicles in IoV can be studied using SI techniques and the networks can be optimized for efficient message communication within the network of vehicles. Additionally, some tools are discussed to help the research community to use those tools, such as OSM, SUMO, and NS3 in traffic scenarios.

There are still some challenges in the efficient routing protocols in IoV and can be focused to improve the message communication during real-time scenarios and use of AI and machine learning techniques can also be incorporated. 

## Figures and Tables

**Figure 1 sensors-23-00555-f001:**
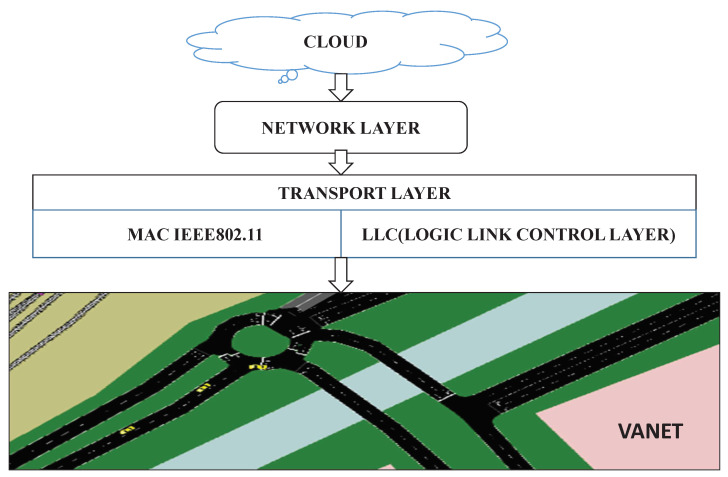
VANET architecture.

**Figure 2 sensors-23-00555-f002:**
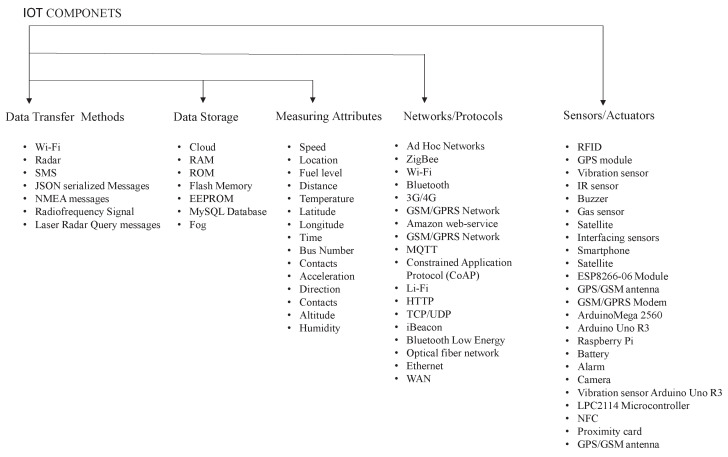
IoT components [4].

**Figure 3 sensors-23-00555-f003:**
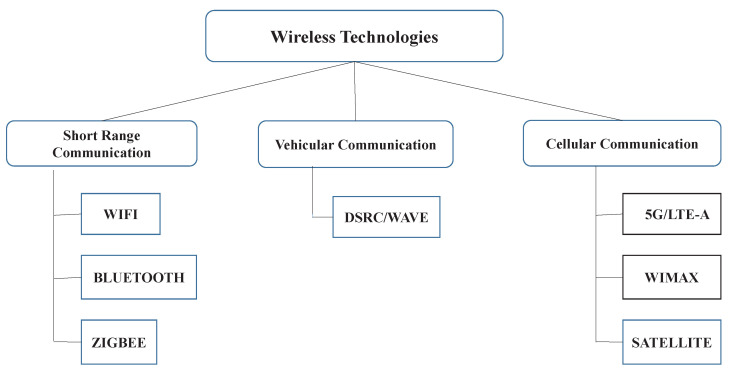
Communication technologies.

**Figure 4 sensors-23-00555-f004:**
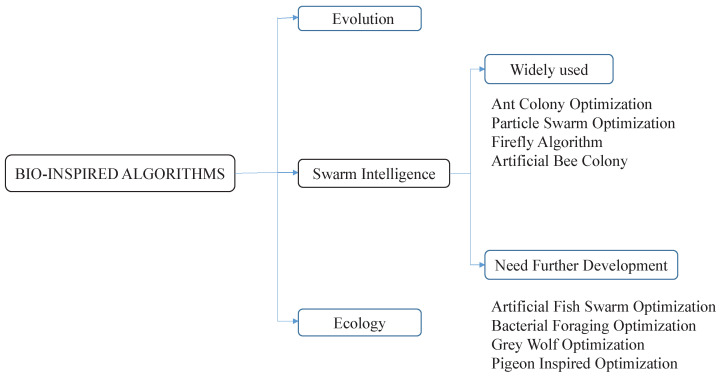
Bio-inspired optimization techniques.

**Figure 5 sensors-23-00555-f005:**
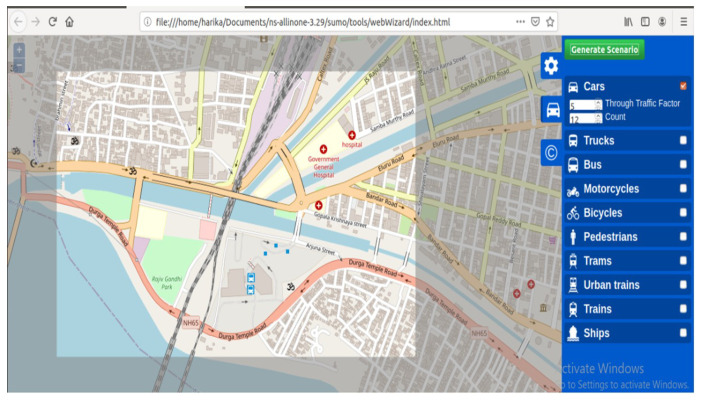
Open street map view.

**Figure 6 sensors-23-00555-f006:**
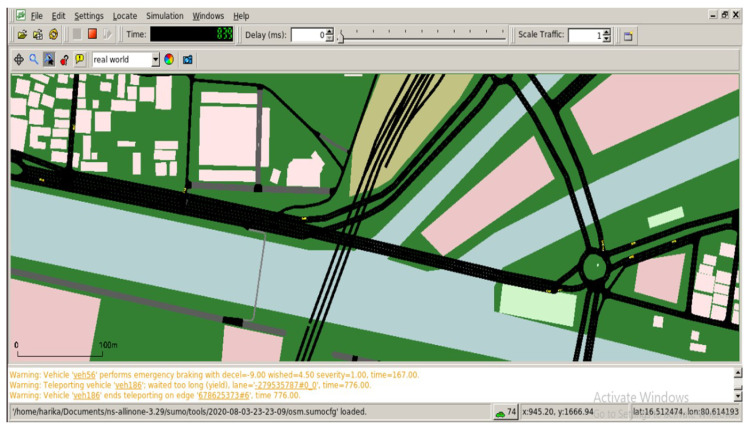
SUMO configuration.

**Figure 7 sensors-23-00555-f007:**
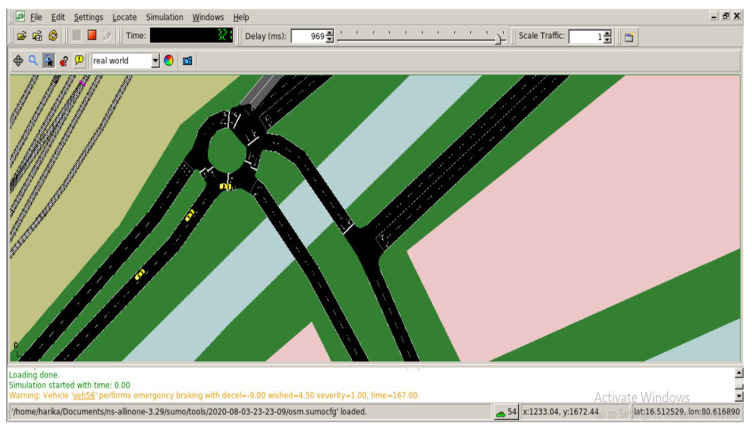
Vehicular traffic in SUMO.

**Figure 8 sensors-23-00555-f008:**
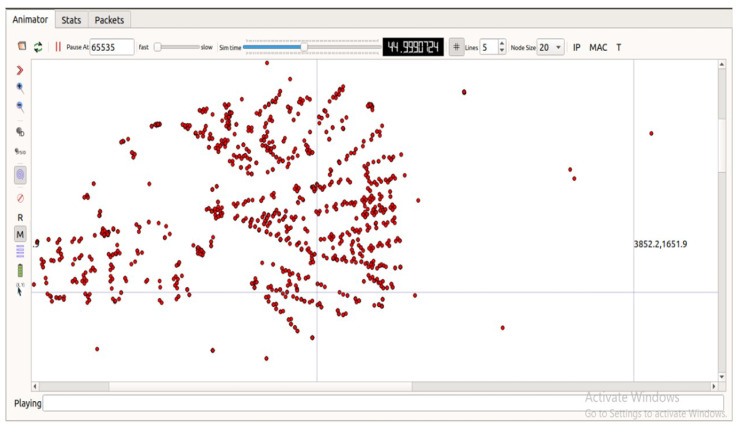
Stats generated in network animator.

**Figure 9 sensors-23-00555-f009:**
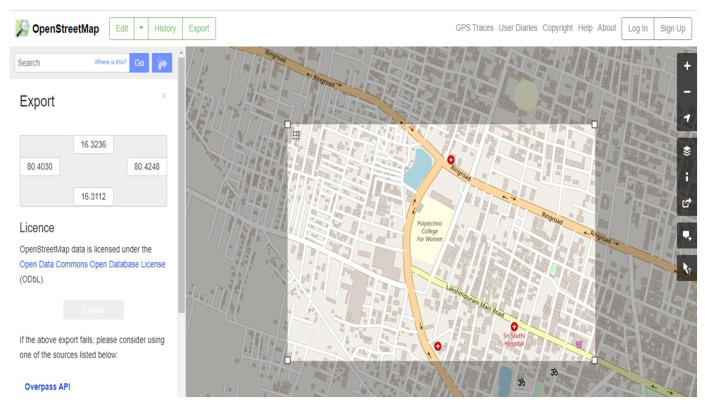
Open street map view for the sample execution.

**Figure 10 sensors-23-00555-f010:**
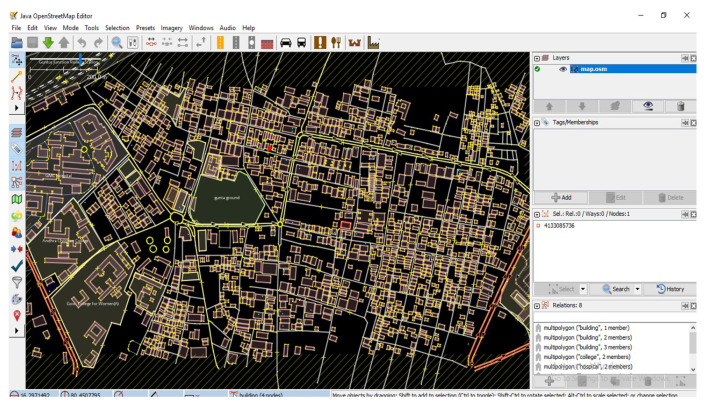
Java open street map editor.

**Figure 11 sensors-23-00555-f011:**
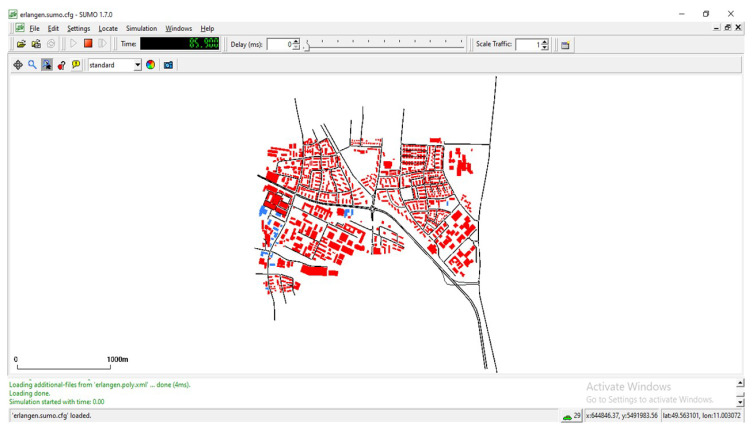
SUMO view.

**Figure 12 sensors-23-00555-f012:**
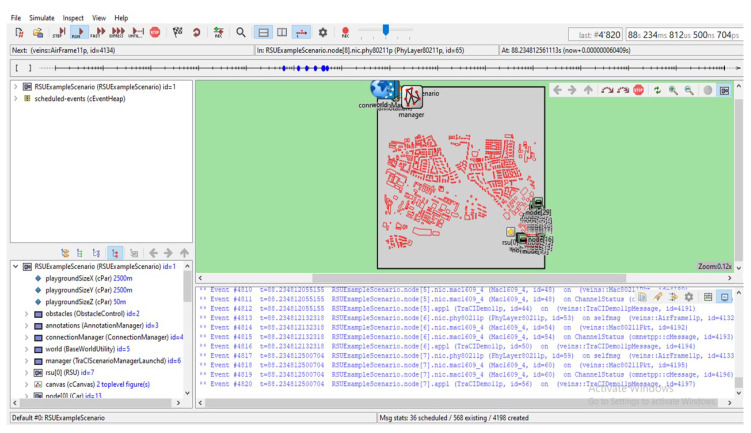
OMnet++ view.

**Figure 13 sensors-23-00555-f013:**
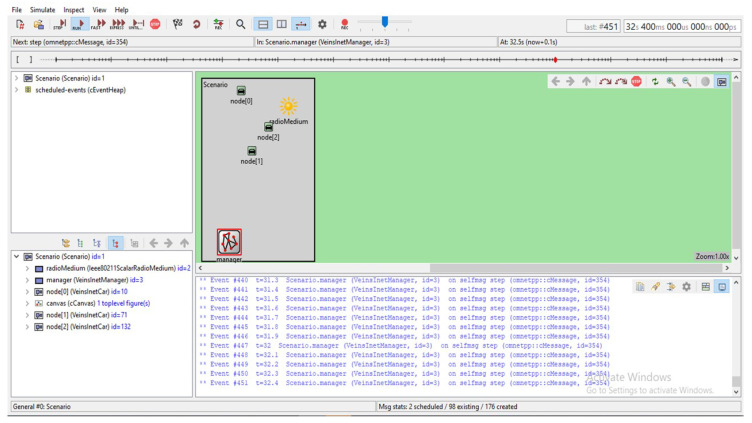
VEINS–INET sample application.

**Table 1 sensors-23-00555-t001:** Applications of IoT.

Safety Applications	Efficient Traffic Management	Support and Infotainment
1. In-Vehicle Signage	1. Road Clog Management	1. Intelligent Parking Route
2. Warning Turn Assistant	2. Toll Management	2. Vehicle Pooling
3. Blind Merge Warning	3. Computerized Map Downloading	3. Web access Provisioning
4. Vehicle Warning	4. Intersection Management	4. Distributed Data Sharing
5. Emergency Electronic Brake Lights	5. SOS Services	5. Clinical Applications
6. Early Detection Warning		
7. Pre-Crash Sensing		
8. Emergency Electronic Brake		

**Table 2 sensors-23-00555-t002:** Swarm based optimization techniques.

Sl.	Algorithm	Publishing Year	Literature
1	Genetic Algorithm	Holland, 1992 [102]	[103,104,105,106]
2	Ant Colony Algorithm	Dorigo and Di Caro, 1999 [97]	[107,108,109]
3	Particle Swarm Optimization	Kennedy and Eberhart, 1995 [110]	[111,112,113]
4	Artificial Bee Colony	Karaboga, 2005 [99]	[114,115,116]
5	Firefly Algorithm	Yang, 2009 [117]	[118,119]
6	Salp Algorithm	Mirjalili, 2017 [100]	[120,121]

**Table 3 sensors-23-00555-t003:** List of research articles and simulation tools used.

Author(s), Year	Reference	Simulation Tools Used
Yang et al., 2013	[163]	MATLAB
Babu et al., 2015	[164]	NS-2, SUMO, and MATLAB
Babu et al., 2016	[165]	OMNET++ and SUMO
Kim et al., 2017	[166]	SUMO and OSM
Abbas et al., 2018	[167]	NS-3 and MATLAB
Lopez et al., 2018	[152]	SUMO and TraCI
Gawas et al., 2019	[168]	OSM, NS-2, VANET MobiSim
Senouci et al., 2019	[169]	NS-2
Shah et al., 2020	[78]	SUMO and MATLAB
Attia et al., 2021	[170]	OMNET++, OSM, and SUMO
Han et al., 2022	[81]	NS-3 and SUMO
Shah et al., 2022	[171]	SUMO and MATLAB

## Data Availability

Not applicable.

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
