# Peer review of "A Survey and Tutorial on Network Optimization for Intelligent Transport System Using the Internet of Vehicles"

_sensors, 2023, doi:10.3390/s23010555_

Round 1
Reviewer 1 Report
The authors examined the current state of different network optimization techniques In IoV. It also emphasizes the context of IoV. This contains applications such as ITS with comparison to other developments, network optimization, IoT debates, and algorithm classification. Some simulation tools are also discussed, which will assist the research community in using such tools for IoV analysis.
The paper appears overall written clearly. It seems well organized, and the used English and language are clear and correct. However, it requires some attention and needs to be elaborated more.
Major comments:
1- In the introduction, the previous surveys that study network optimization using IoV, as well as their contributions, difficulties, and prospects, must be established.
2-Open challenges and future directions must be identified.
3- The conclusion section is not well developed.
Minor comments:
1- I suggest investigating more in other recent studies such :
- Xin, Qin, Mamoun Alazab, Rubén González Crespo, and Carlos Enrique Montenegro-Marin. "AI-based quality of service optimization for multimedia transmission on Internet of Vehicles (IoV) systems." Sustainable Energy Technologies and Assessments 52 (2022): 102055.
- Musa, Salahadin Seid, Marco Zennaro, Mulugeta Libsie, and Ermanno Pietrosemoli. 2022. "Mobility-Aware Proactive Edge Caching Optimization Scheme in Information-Centric IoV Networks" Sensors 22, no. 4: 1387. https://doi.org/10.3390/s22041387
- Rateb Jabbar, Moez Krichen, Mohamed Kharbeche, Noora Fetais, Kamel Barkaoui. A Formal Model-Based Testing Framework for Validating an IoT Solution for Blockchain-based Vehicles Communication. 15th International Conference on Evaluation of Novel Approaches to Software Engineering, May 2020, Prague, Czech Republic. pp.595-602
- Xu, Lingwei, Xinpeng Zhou, Yong Fu, Guanwu Jiang, Xu Yu, Miao Yu, Neeraj Kumar, and Mohsen Guizani. "Accurate and Efficient Performance Prediction for Mobile IoV Networks Using GWO-GR Neural Network." IEEE Internet of Things Journal (2022).
2- The description of the figures is very short. It must be more detailed
3- Luck of references in many areas, such as :
-Line 98: Electric vehicles are not a dream anymore
-Line 141: It joins communication from Machine
-Line 145 : These days, different smart wearable
-Line 226 : Wireless Ad Hoc Networks is a class comprising wireless
- Line 235 : Infrastructure-less MANET is an organization of mobile devices
-Line 269: IoV is viewed to be a development pertaining to V2V network
-Line 429 : The SI algorithm understands the solution for the problem by learning
-Line 479 : The objective of the ants is to locate
4-There exist some minor typos that still need a double-check and correction:
-Example 1: “Section 2, provides a background of IoT and and its role is in ITS.”=> “Section 2, provides a background of IoT and its role is in ITS.”
-Example 2: “new technologies and mechanisms have come up and have been ad-vancing like wireless and sensor technologies, Artificial Intelligence, M2M Communication,Machine Learning, Big Data Analytics, and Neural Networks” =>Please remove the duplication (Machine Learning, Artificial Intelligence and Neural Networks)
Author Response
We the authors thank the reviewer for suggesting revisions for possible inclusion of our manuscript after revision. We tried our best to incorporate the same accordingly.
Major comments:
1- In the introduction, the previous surveys that study network optimization using IoV, as well as their contributions, difficulties, and prospects, must be established.
We have added the text in the introduction accordingly (subsection 1.3: Related Work).
2-Open challenges and future directions must be identified.
There are many open challenges in IoV. We have focused (identified) only on network optimization and have done the survey accordingly.
3- The conclusion section is not well developed.
The conclusion is modified in the revised version.
Minor comments:
1- I suggest investigating more in other recent studies
i. Xin, Qin, Mamoun Alazab, Rubén González Crespo, and Carlos Enrique Montenegro-Marin. "AI-based quality of service optimization for multimedia transmission on Internet of Vehicles (IoV) systems." Sustainable Energy Technologies and Assessments 52 (2022): 102055.
ii. Musa, Salahadin Seid, Marco Zennaro, Mulugeta Libsie, and Ermanno Pietrosemoli. 2022. "Mobility-Aware Proactive Edge Caching Optimization Scheme in Information-Centric IoV Networks" Sensors 22, no. 4: 1387. https://doi.org/10.3390/s22041387
iii. Rateb Jabbar, Moez Krichen, Mohamed Kharbeche, Noora Fetais, Kamel Barkaoui. A Formal Model-Based Testing Framework for Validating an IoT Solution for Blockchain-based Vehicles Communication. 15th International Conference on Evaluation of Novel Approaches to Software Engineering, May 2020, Prague, Czech Republic. pp.595-602
iv. Xu, Lingwei, Xinpeng Zhou, Yong Fu, Guanwu Jiang, Xu Yu, Miao Yu, Neeraj Kumar, and Mohsen Guizani. "Accurate and Efficient Performance Prediction for Mobile IoV Networks Using GWO-GR Neural Network." IEEE Internet of Things Journal (2022).
Within this stipulated time, we couldn't study all but included some most-relevant literature (i and ii) in section 2.3.3. We will study the rest as part of our future work.
2- The description of the figures is very short. It must be more detailed.
The figures (5-13) are the screenshots of the simulation tool demonstration and we think detailed explanation is not required. We hope the explanation is sufficient for the rest of the figures.
3- Lack of references in many areas, such as:
-Line 98: Electric vehicles are not a dream anymore
-Line 141: It joins communication from Machine
-Line 145 : These days, different smart wearable
-Line 226 : Wireless Ad Hoc Networks is a class comprising wireless
- Line 235 : Infrastructure-less MANET is an organization of mobile devices
-Line 269: IoV is viewed to be a development pertaining to V2V network
-Line 429 : The SI algorithm understands the solution for the problem by learning
-Line 479 : The objective of the ants is to locate
As suggested by the reviewer, we have included some of the references accordingly.
4-There exist some minor typos that still need a double-check and correction:
-Example 1: “Section 2, provides a background of IoT and and its role is in ITS.”=> “Section 2, provides a background of IoT and its role is in ITS.”
-Example 2: “new technologies and mechanisms have come up and have been ad-vancing like wireless and sensor technologies, Artificial Intelligence, M2M Communication, Machine Learning, Big Data Analytics, and Neural Networks” =>Please remove the duplication (Machine Learning, Artificial Intelligence and Neural Networks)
We have rectified these, and also verified the whole manuscript again thoroughly and made rectifications.
Reviewer 2 Report
This paper provides a survey and tutorial on Network Optimization for Intelligent Transport System using the Internet of Vehicles. It includes background introduction to IoT and IoV, comparison of IoT and other advancements, discussion of IoT, categorization of algorithms and explanation of simulation tools. The content of this paper is very rich and significant, but there are still some suggestions for improvement.
1. Line 6, I find it hard to agree that the paper does a comprehensive survey of network optimization in the Internet of vehicles. There may be some that it does not cover, such as network evaluation and so on.
2. Line 7, I'm not sure if you're expressing a comparison between ITS and other advancements, or IoT and other advancements.
3. There are some technical terms mentioned, such as M2M, V2V, V2I, etc. Perhaps you can add the full name to help reader understand.
4. Line 147, the paper says "IoT is a subset of IoT", please forgive me for not understanding what you mean.
5. Line 336, the previous article has been talking about IoV, but here the subject is IoT. Do you think IoV and IoT are the same?
6. In 4.2 various network optimization algorithms are introduced, and some cases are also introduced. But I think we're more concerned with the strengths and limitations of these algorithms.
7. Line 647, it says “IoV simulators provide simulation execution without running another software”. However, I think you know that SUMO was also run when using VEINS simulations.
8. In section 5, The tutorial of NS3 and VEINS simulator used in vehicular network simulation is introduced, but the paper only introduces the use of SUMO to output the mobility file of the vehicle, but does not mention the construction or optimization of the network.
9. Figure 5 has the same title with Figure 9. It will make some confuse.
10. In this paper, evolution and bionic optimization algorithms are used to improve the routing of the Internet of vehicles to improve the performance of the Internet of vehicles network. As far as I know, there are many bionic algorithms that have been used in the improvement methods of routing in the Internet of vehicles. What are the similarities and differences between the future research and the previous improvement methods? Perhaps it can be described in more detail.
Author Response
We, the authors thank the reviewer for suggesting revisions for possible inclusion of our manuscript after revision. We tried our best to incorporate the same accordingly.
The content of this paper is very rich and significant, but there are still some suggestions for improvement.
1. Line 6, I find it hard to agree that the paper does a comprehensive survey of network optimization in the Internet of vehicles. There may be some that it does not cover, such as network evaluation and so on.
We agree that some areas we have not considered, hence the text in the abstract is revised accordingly.
2. Line 7, I'm not sure if you're expressing a comparison between ITS and other advancements, or IoT and other advancements.
we have modified the text to make it clear "a comparison between ITS and other advancements"
3. There are some technical terms mentioned, such as M2M, V2V, V2I, etc. Perhaps you can add the full name to help reader understand.
Added accordingly. Thanks for the suggestion.
4. Line 147, the paper says "IoT is a subset of IoT", please forgive me for not understanding what you mean.
Modified accordingly "M2M is a subset of IoT". The sentence is revised accordingly.
5. Line 336, the previous article has been talking about IoV, but here the subject is IoT. Do you think IoV and IoT are the same?
We accept that it is a typographical mistake. The context is IoV only. we have modified in the revised version.
6. In 4.2 various network optimization algorithms are introduced, and some cases are also introduced. But I think we're more concerned with the strengths and limitations of these algorithms.
We do agree and are working as part of the future work.
7. Line 647, it says “IoV simulators provide simulation execution without running another software”. However, I think you know that SUMO was also run when using VEINS simulations.
Yes, we agree. But it was a general statement meant for integrated-frameworks.
8. In section 5, The tutorial of NS3 and VEINS simulator used in vehicular network simulation is introduced, but the paper only introduces the use of SUMO to output the mobility file of the vehicle, but does not mention the construction or optimization of the network.
As the main objective of our research is network optimization, this (construction or optimization of the network) we are considering after literature survey as part of our future work.
9. Figure 5 has the same title with Figure 9. It will make some confuse.
we have revised the caption of Figure 9 accordingly to remove the confusion.
10. In this paper, evolution and bionic optimization algorithms are used to improve the routing of the Internet of vehicles to improve the performance of the Internet of vehicles network. As far as I know, there are many bionic algorithms that have been used in the improvement methods of routing in the Internet of vehicles. What are the similarities and differences between the future research and the previous improvement methods? Perhaps it can be described in more detail.
In the revised version, we have made a comparative analysis of some existing literature and mentioned in the Introduction part (subsection 1.3: Related Work).
Reviewer 3 Report
This paper presents a survey and tutorial on network optimization for intelligent transport system using the internet of vehicles. I have the following concerns:
1. A table should be given in Section 1 (Introduction) presenting novelty with existing works.
2. In section 2.3.1 safety applications are described. The difference between safety and non-safety applications should be given [i].
[i]. A. F. M. S. Shah, M. A. Karabulut, H. Ilhan and U. Tureli, “Performance optimization of cluster-based MAC protocol for VANETs,” in IEEE Access, vol. 8, no. 1, pp. 167731 - 167738, 2020.
3. In Section 3.1, all subsections should be presented with a table describing recent works with their contributions, research gaps, etc. For example, in 3.2, different recent protocols [ii-v] can be mentioned with their contributions, how they evaluate the performance, weaknesses, etc.
[ii]. M. A. Karabulut, A. F. M. S. Shah, and H. Ilhan, “A Novel MIMO-OFDM based MAC Protocol for VANETs,” in IEEE Transactions on Intelligent Transportation Systems, vol. 23, no. 11, pp. 20255-20267, Nov. 2022.
[iii]. J. Wu, H. Lu, Y. Xiang, F. Wang and H. Li, "SATMAC: Self-Adaptive TDMA-Based MAC Protocol for VANETs," in IEEE Transactions on Intelligent Transportation Systems, vol. 23, no. 11, pp. 21712-21728, Nov. 2022.
[iv]. A. F. M. S. Shah, H. Ilhan and U. Tureli, “CB-MAC: A Novel Cluster-Based MAC Protocol for VANETs,” in IET Intelligent Transport Systems, vol. 13, no. 4, pp. 587-595, 2019.
[v]. Han, S.-Y.; Zhang, C.-Y. ASMAC: An Adaptive Slot Access MAC Protocol in Distributed VANET. Electronics 2022, 11, 1145.
4. Different performance metrics such as throughput, delay, channel utilization, etc. should be discussed with mathematical representation.
5. For VANETs, MAC and PHY layers specifications are outlined in the IEEE 802.11 standard, which incorporated the IEEE 802.11p standard. Differences between them and optimization techniques can be presented [v].
[vi]. A. F. M. S. Shah, M. A. Karabulut, H. Ilhan and U. Tureli, "Optimizing Vehicular Safety Message Communications by Adopting Transmission Probability with CW Size," in IEEE Access, vol. 10, pp. 118849-118857, Nov. 2022.
6. In Section 4, a table should be given describing recent works with their contributions, advantages, disadvantages, research gaps, etc.
7. In Section 5, different simulation tools are described. A table should be given presenting recent works with which tools are used, why used, etc.
8. A separate section should be given to highlight future research directions.
9. Please proofread the manuscript. There are a lot of typos.
Author Response
We, the authors thank the reviewer for suggesting revisions for possible inclusion of our manuscript after revision. We tried our best to incorporate the same accordingly.
Comments/Suggestions:
1. A table should be given in Section 1 (Introduction) presenting novelty with existing works.
In the revised version, we have included some existing literature and mentioned in the Introduction part (subsection 1.3: Related Work).
2. In section 2.3.1 safety applications are described. The difference between safety and non-safety applications should be given [i].
[i]. A. F. M. S. Shah, M. A. Karabulut, H. Ilhan and U. Tureli, “Performance optimization of cluster-based MAC protocol for VANETs,” in IEEE Access, vol. 8, no. 1, pp. 167731 - 167738, 2020.
We have included the article for different tools used by research community (in Section 5).
3. In Section 3.1, all subsections should be presented with a table describing recent works with their contributions, research gaps, etc. For example, in 3.2, different recent protocols [ii-v] can be mentioned with their contributions, how they evaluate the performance, weaknesses, etc.
[ii]. M. A. Karabulut, A. F. M. S. Shah, and H. Ilhan, “A Novel MIMO-OFDM based MAC Protocol for VANETs,” in IEEE Transactions on Intelligent Transportation Systems, vol. 23, no. 11, pp. 20255-20267, Nov. 2022.
[iii]. J. Wu, H. Lu, Y. Xiang, F. Wang and H. Li, "SATMAC: Self-Adaptive TDMA-Based MAC Protocol for VANETs," in IEEE Transactions on Intelligent Transportation Systems, vol. 23, no. 11, pp. 21712-21728, Nov. 2022.
[iv]. A. F. M. S. Shah, H. Ilhan and U. Tureli, “CB-MAC: A Novel Cluster-Based MAC Protocol for VANETs,” in IET Intelligent Transport Systems, vol. 13, no. 4, pp. 587-595, 2019.
[v]. Han, S.-Y.; Zhang, C.-Y. ASMAC: An Adaptive Slot Access MAC Protocol in Distributed VANET. Electronics 2022, 11, 1145.
We have included these literature as part of the text in Section 3.2.
4. Different performance metrics such as throughput, delay, channel utilization, etc. should be discussed with mathematical representation.
As this is a survey paper, we thought not to include these general mathematical representations. We hope you understand.
5. For VANETs, MAC and PHY layers specifications are outlined in the IEEE 802.11 standard, which incorporated the IEEE 802.11p standard. Differences between them and optimization techniques can be presented [vi].
[vi]. A. F. M. S. Shah, M. A. Karabulut, H. Ilhan and U. Tureli, "Optimizing Vehicular Safety Message Communications by Adopting Transmission Probability with CW Size," in IEEE Access, vol. 10, pp. 118849-118857, Nov. 2022.
We have included the article for different tools used by research community (Table 3 in Section 5).
6. In Section 4, a table should be given describing recent works with their contributions, advantages, disadvantages, research gaps, etc.
We could not add the table due to some unavoidable circumstances. We will try this as part of our future work.
7. In Section 5, different simulation tools are described. A table should be given presenting recent works with which tools are used, why used, etc.
We have presented a table accordingly (Table 3).
8. A separate section should be given to highlight future research directions.
We have added in the conclusion section.
9. Please proofread the manuscript. There are a lot of typos.
We verified the manuscript again thoroughly and made rectifications.
Round 2
Reviewer 1 Report
The paper has been accepted in its current form, with the authors making the most requested corrections
Reviewer 2 Report
I have no more comments on this revision.
Reviewer 3 Report
I think the authors have addressed my comments well. I have no further comments.